# From Jekyll to Hyde: The Yeast–Hyphal Transition of *Candida albicans*

**DOI:** 10.3390/pathogens10070859

**Published:** 2021-07-07

**Authors:** Eve Wai Ling Chow, Li Mei Pang, Yue Wang

**Affiliations:** 1Institute of Molecular and Cell Biology (IMCB), Agency for Science, Technology and Research (A*STAR), 61 Biopolis Drive, Proteos, Singapore 138673, Singapore; wlechow@imcb.a-star.edu.sg; 2National Dental Centre Singapore, National Dental Research Institute Singapore (NDRIS), 5 Second Hospital Ave, Singapore 168938, Singapore; pang.li.mei@ndcs.com.sg; 3Department of Biochemistry, Yong Loo Lin School of Medicine, National University of Singapore, 10 Medical Drive, Singapore 117597, Singapore

**Keywords:** polymorphism, hyphal morphogenesis, hyphal activation, signal transduction pathways, cell cycle regulation

## Abstract

*Candida albicans* is a major fungal pathogen of humans, accounting for 15% of nosocomial infections with an estimated attributable mortality of 47%. *C. albicans* is usually a benign member of the human microbiome in healthy people. Under constant exposure to highly dynamic environmental cues in diverse host niches, *C. albicans* has successfully evolved to adapt to both commensal and pathogenic lifestyles. The ability of *C. albicans* to undergo a reversible morphological transition from yeast to filamentous forms is a well-established virulent trait. Over the past few decades, a significant amount of research has been carried out to understand the underlying regulatory mechanisms, signaling pathways, and transcription factors that govern the *C. albicans* yeast-to-hyphal transition. This review will summarize our current understanding of well-elucidated signal transduction pathways that activate *C. albicans* hyphal morphogenesis in response to various environmental cues and the cell cycle machinery involved in the subsequent regulation and maintenance of hyphal morphogenesis.

## 1. Introduction

*Candida albicans* is a commensal fungus that is usually a benign member of the microflora in the gastrointestinal tract, genitourinary tract, mouth, and skin of most healthy individuals [1,2,3,4]. *C. albicans* is also an opportunistic fungal pathogen responsible for infections ranging from mild superficial infections to life-threatening candidemia [5]. The use of modern medical therapies such as broad-spectrum antibiotics, cancer chemotherapy, and solid organ transplant has led to an increase in the population vulnerable to invasive candidiasis [6,7]. *C. albicans* is a leading cause of hospital-acquired infections; in the intensive care unit (ICU), candidemia may represent up to 15% of nosocomial infections with an estimated attributable mortality of 47% [7,8,9,10,11].

*C. albicans* displays a wide range of virulence factors and fitness attributes, including its capacity for rapid evolution of resistance to commonly used antifungals (e.g., azoles, polyenes, and echinocandins) and its ability to form biofilms on medical devices, contributing to its success as a pathogen. One striking feature that allows *C. albicans* to cross the commensal-to-pathogen boundary is its ability to switch reversibly between two morphological forms, namely unicellular budding yeast, or filamentous form (hyphae and pseudohyphae), in response to various environmental cues that reflect the host environment [12,13,14,15,16,17,18,19].

Yeast, hyphal, and pseudohyphal forms of *C. albicans* are all present in tissues of human patients and animals with systemic invasive candidiasis [20,21]. Yeast cells exhibit a round-to-oval cell morphology that arises from budding and nuclear division [22]. In contrast, hyphae consist of tubular cells that remain firmly attached following cytokinesis without a constriction at the site of separation. Pseudohyphae share features resembling both yeasts and hyphae, which are branched chains of elongated yeast cells with constrictions at the septum. Both yeast and hyphal forms have crucial and complementary roles important for infection [23]. For instance, the yeast form is required for adhesion to endothelial cells and dissemination into the bloodstream, while the hyphal form is required for tissue penetration during the early stages of infection and yielding resistance towards phagocytosis [24,25,26,27,28,29]. Hyphae-specific virulence factors such as adhesins (Hwp1, Als3, Als10, Fav2, and Pga55), host tissue degrading proteases (Sap4, Sap5, and Sap6), and cytolytic peptide toxin (Ece1), aggrandize the host cell damage during infection [22,30].

Although *C. albicans* can undergo an array of morphological transitions such as the formation of chlamydospores, gray cells, GUT (gastrointestinally induced transition) phenotype, and white/opaque cells, the yeast-to-hyphae transition appears to be the most critical virulence trait [12,15]. Mutants locked in either the hyphal or yeast form have shown diminished virulence, suggesting that the ability to switch between the two morphological forms is essential for virulence [31,32]. Recent advances in mechanistic studies have provided insights into the morphological regulation, coordination, and interplay between environmental factors and genes associated with pathogenesis. This review provides an update on the signal transduction pathways involved in activating *C. albicans* hyphal morphogenesis and how the cell cycle progression and its machinery further aid the regulation and maintenance of sustained hyphal growth.

## 2. Environmental Cues Inducing the Yeast-to-Hyphae Transition

*C. albicans* has adapted to growth in the human host and can transit from yeast to hyphae under a diverse range of environmental cues, as shown in Figure 1 [16,17,19]. Depending upon the cues encountered, morphogenesis can be triggered via several pathways which activate different regulatory circuits.

### 2.1. Host Niches

The human host presents one of the most favorable environments for *C. albicans* morphogenesis due to the presence of multiple inducing factors such as elevated (body) temperature (37 °C), the presence of serum, elevated carbon dioxide (CO_2_) levels (~5%), and low glucose content (0.1%) [33,34,35,36]. Hyphal initiation requires an increase in temperature to 37 °C and the release of quorum-sensing molecules (e.g., farnesol) for the temporary clearance of the major transcription repressor of hyphal morphogenesis Nrg1 [37]. Elevated temperature is also known to promote filamentation through the molecular chaperone Hsp90 and the transcription factor Sfl2 [38,39]. In combination with elevated temperature, the host serum is one of the strongest inducers of *C. albicans* hyphal morphogenesis [40]. Our group has previously demonstrated that bacterial peptidoglycans present in the host serum trigger the hyphal growth of *C. albicans* by directly activating the cyclic AMP (cAMP)-protein kinase A (PKA) signaling cascades through adenylyl cyclase Cyr1 [35,41]. Similarly, CO_2_, another potent inducer of filamentous growth, is also known to activate the cAMP-PKA pathway by binding to Cyr1 [42]. In *C. albicans*, a carbonic anhydrase Nce3 is involved in CO_2_ signaling and conversion of CO_2_ to bicarbonate (HCO_3_^-^) [33]. Especially in host niches with limited CO_2_ (e.g., on the skin), the CO_2_/HCO_3_^-^ equilibration controlled by Nce3 is crucial for the pathogenesis of *C. albicans*. The G-protein-coupled receptor Grp1 and the Gα protein Gpa2 act as the glucose-sensing network for *C. albicans* morphogenesis [43]. Low glucose concentration present in the bloodstream results in the maximal hyphal formation, while high glucose concentrations repress it [36]. The factors mentioned above have been shown to activate the fungal cAMP-PKA signaling pathway [33,44,45,46].

### 2.2. Hypoxia (Low Oxygen)

Hypoxia is a clinical characteristic of inflammatory conditions, representing zones of intense immune activity [47,48]. *C. albicans* can modulate the host response under hypoxia and anoxia (absence of oxygen) to evade immune responses [47]. As a commensal, *C. albicans* adapts to hypoxia condition by repressing the transcription factor of filamentous growth Efg1. Interestingly, Efg1 has a dual role in hyphal morphogenesis. Under hypoxia, it acts as a repressor at temperatures ≤ 35 °C, while under normoxia (normal oxygen level), Efg1 is a strong inducer of hyphal formation [48]. *efg1*Δ/Δ mutants displayed hyperfilamentous growth at temperatures ≤ 35 °C during hypoxic growth on agar surface or during embedment in agar but not during growth in liquid media [48,49]. In contrast, Ace2 is essential for hyphal morphogenesis under hypoxia while being dispensable under normoxia [50,51]. Efg1 and Ace2 share functional overlap; chromatin immunoprecipitation on microchips (ChIP) analyses revealed that hypoxic repressors (Efg1 and Bcr1) and hypoxic activators (Ace2 and Brg1) are connected in regulatory circuits in controlling hyphal morphogenesis under hypoxia conditions [48]. Additionally, Efg1 was implicated in the Cek1-mediated pathway under hypoxia at ≤35 °C; low Efg1 phosphorylation levels inhibit Cek1 and Cph1, preventing hyphal morphogenesis. The low Efg1 phosphorylation levels also inhibited hyphal morphogenesis through the cAMP-PKA pathway.

### 2.3. pH Conditions

*C. albicans* is constantly exposed to fluctuations in pH ranging from acidic to slightly alkaline in different human body niches such as the digestive tract, vagina, oral cavity, blood, and tissues [52]. pH sensing is mediated through Rim101, an important regulator of the yeast-to-hyphae morphological transition [53,54,55]. Upon activation, the transcription factor Rim101 enters the nucleus and mediates pH-dependent responses [56]. Remarkably, *C. albicans* is not only capable of sensing and adapting to environmental pH but can also modulate extracellular pH by alkalinizing its surrounding environment and auto-inducing hyphal formation [57]. Furthermore, alkalinization has been shown to counter the macrophage acidification during engulfment, promoting its survival in the macrophage [58].

### 2.4. N-Acetylglucosamine (GlcNAc)

GlcNAc is commonly found as a structural component of the mucosa of the gastrointestinal tract, bacterial cell wall peptidoglycan, and fungal cell wall chitin [59,60]. Given the ubiquitous nature of GlcNAc in host niches and microbial cells, it could potentially serve as a critical signaling molecule that regulates the switch between the commensalism and pathogenicity of *C. albicans* [61]. Figure 2 depicts an update on GlcNAc signaling pathways and their involvement in hyphal morphogenesis. Ngt1 was identified as a membrane transporter specific for GlcNAc, indicating the importance of GlcNAc in intracellular signaling [62]. However, metabolism or breakdown of GlcNAc intracellularly is not required in the *C. albicans* hyphal morphogenesis as triple deletion mutants that lack all three catabolic genes (*HXK1*, *NAG1*, and *DAC1*) can exhibit filamentous growth with the addition of exogenous GlcNAc [63]. Interestingly, genetic screens have revealed two novel transcription factors, *NGS1* and *RON1*, which play essential roles in both the GlcNAc catabolism and GlcNAc-induced filamentous growth [64]. *NGS1* encodes a protein similar to the GNAT family of histone acetyltransferase Gcn5, while *RON1* encodes a protein similar to the Ndt80-like DNA-binding domain [65]. Ngs1 was discovered as a novel GlcNAc signal sensor and transducer for GlcNAc-induced transcription in *C. albicans* [65]. Ngs1 targets the promoters of GlcNAc-inducible genes constitutively via the transcription factor Rep1 [65]. Ron1 was initially thought to act as both an activator and a repressor of hyphal morphogenesis. However, *ron1*Δ/Δ mutants constructed using the CRISPR/Cas9 method did not display observable GlcNAc-induced filamentous growth [64,66]. It is noteworthy that, upon the addition of GlcNAc, *ndt80 ron1* double deletion mutants could overcome the hyphal defects observed in *ndt80*Δ/Δ mutants. Collectively, it suggests that Ron1 functions as a repressor of filamentous growth in the absence of Ndt80 [66]. The GlcNAc signaling pathway was initially believed to be related to the cAMP-PKA pathway as *cyr1*Δ/Δ mutants cannot form hyphae under a broad range of conditions, including GlcNAc [67]. However, it was later discovered that the fast-growing *cyr1* pseudo revertant strains could undergo filamentous growth in a GlcNAc containing medium [68]. This indicates that GlcNAc can stimulate a signaling pathway independent of the cAMP-PKA pathway that has yet to be fully elucidated. An alternative pathway involved in the GlcNAc signaling is the pH-sensing Rim101 pathway. Production of excess ammonia during GlcNAc catabolism results in an increase in extracellular pH (>5), which indirectly stimulates the hyphal induction in *C. albicans* via the Rim101 pathway [60,69].

### 2.5. Amino Acids Sensing

*C. albicans* can utilize amino acids as alternative carbon sources during growth in glucose-poor, amino acid-rich conditions [57]. Amino acids that can be catabolized to arginine and proline are potent inducers of hyphal morphogenesis [70,71,72]. *C. albicans* detects extracellular amino acids via the plasma membrane-localized SPS (Ssy1-Ptr3-Ssy5) complex, which regulates two paralogous transcription factors, Stp1 and Stp2 (Figure 2) [73,74,75]. In the presence of extracellular amino acids, the amino acid sensor Ssy1 (Cys1) activates amino acid permease (AAP) genes [76], while the peripherally membrane-associated Ptr3 recruits casein kinase I (CKI), which activates the endoproteolytic activity of the endoprotease Ssy5 [76,77]. Ssy5 endoproteolytically cleaves the nuclear exclusion domain of Stp1 and Stp2, facilitating their translocation to the nucleus [74]. Processed Stp1 regulates the expression of *SAP2*, which encodes the major secreted aspartyl proteinase, and *OPT1*, which encodes an oligopeptide transporter. The active Stp2 activates the expression of a subset of AAP genes [74,75]. The endoplasmic reticulum (ER) chaperone protein Csh3 is required for the proper expression and plasma membrane localization of Ssy1 and AAPs [78]. *ssy1*Δ/Δ, *ptr3*Δ/Δ, *ssy5*Δ/Δ, *csh3*Δ/Δ, and *stp2*Δ/Δ mutants fail to respond to the presence of extracellular amino acids and display impaired filamentous growth [73,74,75,76,78]. Amino acid-induced morphogenesis has recently been shown to be dependent on proline catabolism, with a strict requirement for Ras1 activity [79]. Proline catabolism in the mitochondria leads to elevated cellular ATP levels, which exceed the critical threshold of ATP needed to induce cAMP synthesis, leading to hyphal morphogenesis [71,79,80].

### 2.6. Quorum Sensing

In addition to host environmental cues, *C. albicans* morphogenesis is also regulated by several endogenous and exogenous quorum-sensing molecules (QSMs) [81,82,83,84,85]. Tyrosol and farnesol are well-known QSMs produced by *C. albicans*, which accelerate and inhibit the yeast-to-hyphae transition, respectively [81,86,87]. *C. albicans* also produces aliphatic alcohols (e.g., ethyl alcohol, isoamyl alcohol, 1-dodecanol, 2-dodecanol, and nerolidol) and aromatic alcohols (e.g., 2-phenylethyl alcohol and tryptophol) that inhibit filamentation and subsequent biofilm formation [88,89]. Farnesol and 1-dodecanol were implicated in the Ras1-cAMP signaling pathway, and hyphal defects can be restored upon the addition of dibutyryl-cAMP [89]. Farnesol also inhibits filamentous growth through the negative regulators Tup1 and Nrg1 [90]. The hyphal morphogenesis of *C. albicans* can also be regulated by interaction with other microorganisms found in the host environment [91,92]. For instance, the coexistence of *C. albicans* and Gram-negative bacteria, such as *Pseudomonas aeruginosa*, *Stenotrophomonas maltophilia*, and *Burkholderia cenocepacia*, is commonly found as mixed infections in the lungs of cystic fibrosis (CF) patients [93]. Exogenous QSMs, namely, 3-oxo-C12-homoserine lactone and phenazines (pyocyanin, phenazine methosulfate, and phenazine-1-carboxylate) secreted by *P. aeruginosa*, were found to inhibit the hyphal development of *C. albicans* [82,94]. Diffusible signal factor (DSF), representing a new class of widely conserved quorum-sensing signals from Gram-negative bacteria, has been implicated in inter-kingdom signaling between *C. albicans* and bacteria [95]. DSF (*cis*-11-methyl-2-dodecenoic acid) produced by *S. maltophilia*, and BDSF (*cis*-2-dodecenoic acid) produced by *B. cenocepacia* play a role in the yeast-to-hyphae transition [95]. DSF released by *S. maltophilia* has been reported to interfere with two key virulence factors of *C. albicans*: the yeast-to-hyphae transition and biofilm formation [96]. Recent microarray studies revealed the involvement of repressors (Ubi4 and Sfl1) and the activator (Sfl2) of filamentous growth in BDSF regulation of hyphal morphogenesis [97]. With the addition of BDSF, elevated levels of Ubi4 and Sfl1 and degradation of Sfl2 block the yeast-to-hyphae transition. *C. albicans* is also commonly found along with other microorganisms in inter-kingdom biofilms [98]. Many bacteria and fungi can secrete glucanases into the environment that digest glucan, the most abundant fungal cell wall component [99,100]. *C. albicans* itself secretes at least three glucanases (Xog1, Exg2, and Spr1) which are involved in cell wall remodeling during cell division and morphogenesis [101,102]. Interestingly, it has been found that β-1,3-glucanase, secreted by bacteria and fungi, can induce filamentous growth in *C. albicans* even at low temperatures (22 °C), in a cell density-dependent manner [103]. *cek1*Δ/Δ and *efg1*Δ/Δ mutants cannot form hyphae in response to β-1,3-glucanase, suggesting that the Cek1-mediated pathway is involved [103].

### 2.7. In Vitro Conditions

Hyphal growth can also be induced using synthetic growth media such as Lee’s medium (pH 7), spider medium, and mammalian tissue culture M199 under laboratory conditions [104,105,106]. Nitrogen starvation-induced filamentation occurs in the low nitrogen SLAD medium via ammonium permease Mep2 sensing [107,108]. Methionine, an amino acid in Lee’s medium, has been reported as the main inducer of yeast-to-hyphae transition via G-protein-coupled receptor Gpr1 sensing [36]. Recently, the methionine permease Mup1 and the S-adenosylmethionine decarboxylase Spe2 were discovered to be crucial for cAMP production in response to methionine [109]. Both nitrogen and amino acid catabolism activate hyphal morphogenesis via the cAMP-PKA pathway.

## 3. The cAMP-PKA Pathway

The cyclic adenosine monophosphate (cAMP)-protein kinase A (PKA) pathway is highly conserved in eukaryotes and regulates many cellular processes in *C. albicans* [44,110]. This pathway plays a critical role in morphogenesis, positively regulating filamentation [111,112,113]. One of the well-studied regulatory targets of the cAMP-PKA pathway is the transcription factor Efg1, which stimulates the expression of numerous hyphal-specific genes through the activation of the transcription factor Ume6 (Figure 3).

The cAMP-PKA pathway is triggered either directly through the adenylyl cyclase Cyr1 or via the small GTPase Ras1, which activates Cyr1, depending upon the stimuli encountered (Figure 3) [114]. Cyr1, in direct association with Cap1 (cyclase-associated protein), drives the conversion of ATP to cAMP [46]. PKA holoenzyme is activated upon cAMP binding to the homodimer regulatory subunit Bcy1, inducing a conformational change releasing the two catalytic subunits, Tpk1 and Tpk2, which then activate downstream target proteins or genes through phosphorylation or the binding of promoter regions to induce transcription (Figure 3) [110,115,116,117]. Tpk1 and Tpk2 are partially redundant. Tpk1 is required for hyphal formation on solid media, whereas Tpk2 is needed for hyphal formation in liquid media and invasive growth into solid media [116]. While the loss of either subunit does not block filamentation, loss of both Tpk1 and Tpk2 completely blocks filamentation [39,116]. cAMP levels are negatively regulated by Pde1 (low-affinity phosphodiesterase) and Pde2 (high-affinity cAMP phosphodiesterase), which increase the rate of cAMP degradation [118,119]. Loss-of-function mutations or deletion of *PDE2* increase cAMP levels, leading to constitutive activation of the pathway and hyper-filamentation [119,120]. The *pde2*Δ/Δ mutants exhibit reduced virulence due to reduced adhesion capability [121]. On the contrary, the *pde1*Δ/Δ mutants can still undergo filamentation [122]. Interestingly, *pde1 pde2* double deletion mutants exhibit attenuated virulence as compared to *pde2*Δ/Δ mutants [121].

The adenylyl cyclase Cyr1 is required for hyphal development and virulence but is not essential for basal growth in *C. albicans* [67]. Deletion of *CYR1* has a global impact on gene expression, resulting in many alterations in response to environmental cues [67,123]. Cyr1 contains several highly conserved functional domains, which include a Gα domain, a Ras-association (RA) domain, a leucine-rich repeat (LRR) domain, a cyclase catalytic (CYCc) domain, and a Cap1 (cyclase-associated protein 1) binding domain (CBD) [124,125].

The small GTPase Ras1, an upstream activator of Cyr1, transduces extracellular signals (serum in combination with elevated temperature or nitrogen starvation) to Cyr1 [44,107,114]. Ras1 usually exists in the cell in an inactive (GDP-bound) form, and its switch to the active form (GTP-bound) is regulated by the GTPase-activating protein (GAP) Ira2; the guanine nucleotide exchange factor (GEF) Cdc25 drives the GTP-Ras1-to-GDP-Ras1 switch (Figure 3) [126]. Active Ras1 directly interacts with Cyr1 through the RA domain, stimulating cAMP production [44,114]. Cyr1 activity depends upon the binding of Cap1 at the CBD domain and the binding of G-actin to Cap1 to form a tripartite complex, which serves to maintain the activation of the pathway [46,127]. Deletion of *CAP1* results in lowered cAMP levels and blocks in morphogenesis.

The presence of serum drives morphogenesis via Ras1 activation of the cAMP-PKA pathway. Deletion of *RAS1* impairs serum-induced filamentous growth, which can be overcome by supplementation with cAMP, and overexpression of cAMP signaling components rescues its defects [128,129]. The serum contains various active factors, such as glucose and bacterial peptidoglycan fragments, that can stimulate the pathway. The Gα domain of Cyr1 is the binding site for a heterotrimeric G-protein α subunit Gpa2, which is activated by the G-protein-coupled receptor Gpr1 in response to glucose and amino acids [34,130]. However, neither Gpr1 nor Gpa2 is required for serum-induced hyphal formation in liquid media [43]. Glucose-induced activation of cAMP synthesis appears to be mediated by Cdc25-Ras1 interaction and not Gpr1 binding of Cyr1 [36,43,45]. The LRR domain of Cyr1 recognizes and binds muramyl dipeptides (MDP), subunits of bacterial peptidoglycan present in serum [35,39,41]. Deletion or mutation of the LRR domain abolishes cAMP-PKA activation in the presence of MDPs [35]. CO_2_ or HCO_3_^-^ bind to the CYCc domain, stimulating the production of cAMP required for hyphal growth [33]. Both endogenous and exogenous QSMs farnesol and 3-oxo-C12-homoserine lactone (HSL) block the hyphal growth by binding to the CYCc domain and inhibiting the activity of Cyr1 [82].

Temperature-dependent morphogenesis via the cAMP-PKA pathway is governed by the heat shock chaperone protein, Hsp90, whose expression is regulated by the heat shock transcription factor, Hsf1 [131]. Under basal conditions, Hsp90 and its co-chaperone Sgt2 interact with Cyr1 and repress it [39,132]. Temperature elevation results in cellular stress leading to an increase in competing Hsp90 client proteins, thereby relieving Hsp90 repression of Cyr1. Inhibition of Hsp90 leads to filamentous growth under non-inducing conditions.

Cell cycle perturbation also induces morphogenesis via the Ras-cAMP-PKA signaling pathway. Disrupting cell cycle progression by treating with the DNA synthesis inhibitor hydroxyurea (HU) arrests cells in the S phase, while prolonged depletion of Cln3 arrests cells in the G1 phase [133,134]. Arrested cells switch to filamentous growth and re-enter the cell cycle via Ras1-activation of the cAMP-PKA pathway [133,134]. However, while cell cycle arrest in G1 and S phase induces morphogenesis, different mechanisms are involved. HU-induced filamentation does not require the downstream transcription factor Efg1, and the hyphal-specific transcription factor Ume6 and the G1 cyclin Hgc1 but involves other PKA target genes [133,135]. In contrast, filamentation due to Cln3 depletion requires Efg1, Ume6, and Hgc1 [133,134,135].

## 4. The MAPK Pathways

The mitogen-activated protein kinase (MAPK) signal transduction pathway consists of three components: the MAP kinase kinase kinase (MAPKKK), the MAP kinase kinase (MAPKK), and the MAP kinase (MAPK) (Figure 3). MAPK signaling is dependent on three phosphotransfer steps. Upon activation, MAPKKK becomes phosphorylated and triggers the phosphorylation of the MAPKK, which in turn phosphorylates the MAPK [136,137]. In *C. albicans*, the Cek1-mediated MAPK pathway and the PKC MAPK pathway are activated by different stimuli. They serve as patterns of cascades that are essential for its morphogenesis and virulence, as shown in Figure 3 [137,138]. Apart from the cAMP-PKA signaling pathway, Ras1 also signals through the MAPK signaling cascade (Cek1-mediated) to coordinate filamentation in response to nitrogen starvation conditions via the Mep2 sensor [107].

### 4.1. The Cek1-Mediated MAPK Pathway

*C. albicans* extracellular signal-regulated kinase (ERK)-like 1 (Cek1)-mediated MAPK pathway is involved in cell wall biogenesis and virulence [139,140]. The Cek1-mediated MAPK pathway also plays an important role in hyphal development through the activation of downstream transcription factor Cph1, a positive regulator of filamentation [106,141]. This pathway can be induced by several factors such as low nitrogen, cell wall damage, osmotic stress, and embedding matrix. Under nitrogen starvation conditions, this pathway is activated by the ammonium permease Mep2 via a Ras1-dependent manner [107]. Cdc42, an essential GTPase, and its GEF Cdc24 are required for filamentous growth and virulence [142,143]. Upon interactions with Cdc24 and Ras1, activated Cdc42 turns on downstream effectors, including p21-activated kinase (PAK) Cst20 and Cla4, which then triggers concerted phosphorylation of the Ste11 (MAPKKK), Hst7 (MAPKK), and Cek1 (MAPK) (Figure 3) [138]. Mutations in the Cek1-mediated cascade cause defects in hyphal development to a different degree under certain conditions and result in attenuated virulence in animal models [144,145]. The Cek1-mediated MAPK pathway can also be activated through its upstream transmembrane proteins via cell wall damage or osmotic stress. Sho1, Opy2, and Msb2 form a complex that interacts with Cdc42 and Cst20, triggering Cek1 phosphorylation [144]. *sho1*Δ/Δ, *opy2*Δ/Δ, and *msb2*Δ/Δ mutants display altered sensitivity to cell wall damaging agents such as Congo Red, zymoylase, and tunicamycin, suggesting their roles in cell wall biogenesis [146,147]. *sho1*Δ/Δ mutants are sensitive to osmotic stress (i.e., 1 M sodium chloride), suggesting its additional role in osmotic stress signaling. The Cek1-mediated MAPK pathway responds to embedded matrix conditions by initiating a signaling cascade that ultimately activates Cph1 via Cek1 [22]. Rac1, Lmo1, and its exchange factor Dck1 are essential for invasive filamentous growth in the embedding matrix [148,149]. In contrast to Cdc42, Rac1 is not required for serum-induced hyphal growth [150]. *rac1*Δ/Δ, *lmo1*Δ/Δ, and *dck1*Δ/Δ mutants were observed to exhibit filamentous growth defects on solid agar and increased sensitivity to cell wall damaging agents, such as Calcofluor White and Congo Red [148]. Intriguingly, the overexpression of the Cek1 MAP kinase in *rac1*Δ/Δ, *lmo1*Δ/Δ, and *dck1*Δ/Δ mutants restores invasive filamentous growth on solid media, suggesting that Rac1, Lmo1, and Dck1 function together upstream of the Cek1-mediated MAPK pathway during invasive filamentous growth [148]. The downstream transcription factor, Cph1, is essential for hyphal growth on solid agar but not in liquid media [106]. It was found that *efg1 cph1* double deletion mutants cannot form filaments under hypha-inducing conditions and are avirulent in animal models [31]. However, *efg1 cph1* double deletion mutants can form filamentation when embedded in the matrix, suggesting the involvement of other transcription factors for hyphal development under this condition [151].

### 4.2. The PKC MAPK Pathway

The protein kinase C (PKC) MAPK pathway is commonly known as the cell wall integrity pathway [137]. Pkc1 activation leads to a MAPK cascade activation of Bck1 (MAPKKK), Mkk1 (MAPKK), and Mkc1 (MAPK). Cellular morphogenesis in *C. albicans* is a highly dynamic process controlled by a master regulator, Rho1, in response to various stressors (Figure 3) [152]. Rho1, the master regulator of the cell wall integrity signaling cascade, is activated by the GEF Rom2 and inactivated by the GAP Lrg1 [153]. Recently, the PKC MAPK pathway was discovered to regulate *C. albicans* morphogenesis through the co-regulation of cAMP signaling [154]. Interestingly, Rho1 plays an important role in filamentation through Pkc1. Pkc1 was found to be a global regulator of *C. albicans* morphogenesis through the regulation of adenylyl cyclase Cyr1. A reduction of Cyr1 activity was observed in *pkc1*Δ/Δ mutants [154]. Lrg1 deactivates Rho1 by locking it in its inactive form, which suppresses the yeast-to-hyphae transition. *C. albicans* morphogenesis is independent of its canonical MAPK cascade. Deletion of *BCK1* or *MKC1* does not impair the filamentous growth in response to the Hsp90 inhibitor geldanamycin or serum [155]. Although the downstream transcription factors of Mkc1 have previously been proposed to be Efg1, Czf1, and Bcr1, to date, *C. albicans* morphogenesis through distinct effector(s) remains elusive [154,155].

## 5. Negative Regulators of Hyphal Morphogenesis

*C. albicans* morphogenesis is negatively regulated by the transcriptional repressors Tup1, Nrg1, and Rfg1 [156,157,158]. Tup1 is a global transcriptional repressor, and its inactivation leads to constitutive filamentous growth and derepression of hyphal-specific genes [130,156,159]. Nrg1 and Rfg1 are well characterized DNA-binding proteins, which regulate different subsets of hyphal-specific genes by recruiting co-repressor Tup1. A DNA microarray analysis revealed significant up-regulation of 61 genes in response to serum and 37 °C [160]. Approximately half of these genes are found to be repressed by the transcription factors Tup1, Nrg1, and Rfg1, suggesting their importance in repressing hyphal morphogenesis. *C. albicans* cells that lack these repressors develop into pseudohyphae with the expression of hyphal-specific genes [161]. Surprisingly, only *nrg1*Δ/Δ mutants form hyphae in response to serum. In addition, *nrg1*Δ/Δ mutants appear to display stronger hyphal phenotypes than *rfg1*Δ/Δ mutants, suggesting its predominant role in the negative regulation of hyphal growth [162].

### 5.1. The Farnesol-Mediated Inhibition Pathway

Though Tup1 is found to act independently of the cAMP-PKA and MAPK pathways to regulate morphogenesis, it seems to play a crucial role in the farnesol response pathway [90,159]. Farnesol, an endogenous QSM, is produced when the cell densities of *C. albicans* are high. While farnesol can block the yeast-to-hyphae transition, it cannot block the elongation of pre-existing filaments [163,164,165]. Morphological and transcriptional studies, which investigated the possible functional overlap between farnesol and hyphal transcriptional repressors, have demonstrated the direct involvement of Tup1 in the farnesol-mediated inhibition of filamentous growth [90]. *tup1*Δ/Δ and *nrg1*Δ/Δ mutants display elevated levels of farnesol and are constitutively filamentous even in the presence of exogenous farnesol. In the presence of farnesol, *TUP1* levels increase, but *NRG1* and *RFG1* levels are unaffected [90]. Further targeted studies on the farnesol-mediated inhibition pathway have unraveled its dedicated mechanistic control of filamentous growth [166]. Upon inoculation of cells, where farnesol inhibition is relieved, the transcriptional repressor Cup9 is constantly degraded by the N-end rule E3 ubiquitin ligase Ubr1, allowing the expression of kinase Sok1 and subsequent degradation of Nrg1. In contrast, the presence of farnesol inhibits the degradation of Cup9, thereby repressing Sok1 expression, which in turn blocks the degradation of Nrg1 and hyphal development [166].

### 5.2. The Roles of Negative Regulators Tup1 and Nrg1 in Hyphal Elongation

Critical regulators of hyphal initiation and the activation of hypha-associated genes, such as Efg1, Cph1, Czf1, and Flo8, are shown in Figure 4. Thereafter, a second regulatory network is required for the hyphal elongation process and long-term maintenance of hyphal growth through Hgc1, Eed1, and Ume6, which are negatively regulated by Tup1 and Nrg1 [144,167,168,169]. Eed1 was first identified in oral tissue infections from patients suffering from oral disease, and its associated regulatory network was explored through comprehensive transcriptomics analysis [167]. Eed1 is positively regulated by Efg1 as the overexpression of *EED1* partially rescues the hyphal defects in *efg1*Δ/Δ mutants. *EED1* expression is significantly up-regulated in the continuously filamentous *nrg1*Δ/Δ and *tup1*Δ/Δ mutants under non-hyphae-inducing conditions [167]. In contrast, under hyphae-inducing conditions, *EED1* levels were slightly decreased in *nrg1*Δ/Δ and *tup1*Δ/Δ mutants, but elevated 10-fold in wild-type cells. Collectively, this suggests that *EED1* is repressed by both Nrg1 and Tup1 in wild-type *C. albicans*. Ume6 acts downstream of Eed1 as the overexpression of *UME6* restored the hyphal elongation defect observed in *eed1*Δ/Δ mutants [167]. *UME6* expression levels were significantly down-regulation in *eed1*Δ/Δ mutants [167]. *HGC1* expression is detected within 5 min of hyphal induction, whereas *UME6* expression is only detected after 15 min upon induction [170]. This suggests that a Ume6-independent mechanism initially induces HGC1. Nrg1 and Tup1 negatively regulate both Ume6 and Hgc1 [161,169]. Ume6 could also be induced as a result of relief of transcriptional repression by the Nrg1-Tup1 complex.

### 5.3. O_2_ and CO_2_ Signaling Pathways for Sustained Hyphal Development

The stability of hyphae-specific transcription factor Ume6 is governed by two parallel pathways in response to O_2_ and CO_2_ concentrations [171,172]. Ofd1 negatively regulates the stability of Ume6 by E3 ubiquitin ligase Ubr1 under hypoxia conditions. *ofd1*Δ/Δ and *ubr1*Δ/Δ mutants can maintain hyphal elongation in atmospheric O_2_ and 5% CO_2_ [171,172]. However, deletion of *UBR1* does not block Ume6 degradation in atmospheric CO_2_, suggesting the involvement of additional E3 ubiquitin ligase in response to CO_2_ [172]. Recently, it was discovered that CO_2_, an inducer of filamentous growth, also plays a critical role in the sustenance of hyphal growth in response to high CO_2_ (5%) [172]. In the CO_2_ signaling of sustained hyphal growth, a type 2C protein phosphatase (PP2C) Ptc2 and a cyclin-dependent kinase Ssn3 were identified to be the major positive and negative regulators, respectively [172]. High CO_2_ induces Ptc2-mediated dephosphorylation of Ssn3. Consequently, the hypophosphorylated Ssn3 fails to phosphorylate Ume6 at the S437 residue. This prevents subsequent ubiquitination of Ume6 by the E3 ubiquitin ligase SCF^Grr1^, resulting in stabilization of Ume6 for the sustenance of hyphal growth.

### 5.4. Negative Regulators as Potential Drug Targets

Recent discoveries have introduced novel compounds that inhibit *C. albicans* hyphal and biofilm formation through the up-regulation of negative regulators Tup1 and Nrg1 [173,174]. Treatment of *C. albicans* with novel synthetic SR analogues, 5-[3-substituted-4-(4-substituted benzyloxy)-benzylidene]-2-thioxo-thiazolidin-4-one derivatives, resulted in a 3 to 4-fold increase in the expression of *TUP1* and a 2-fold increase in the expression of *NRG1*, which effectively inhibits the hyphal morphogenesis [173]. Copper oxide nanoparticle (Cu_2_O-NP) was found to inhibit the yeast-to-hyphae transition through the down-regulation of *RAS1* and up-regulation of *NRG1* and *TUP1* [174]. Exploiting the negative regulators as drug targets holds excellent potential for future clinical applications. There is a growing interest in applying nanoparticles on medical devices, prosthetic devices, and catheters to combat polymicrobial biofilms in clinical settings.

## 6. Mechanisms of Hyphal Morphogenesis

### 6.1. Septin Ring Formation

Although the septin subunits are static in budding yeast cells, upon hyphal induction, Cdc3, Cdc12, and Sep7 form a stable core, while the Cdc10 subunit becomes dynamic, shuttling between the septin ring and the cytoplasm [175]. Cdc3 and Cdc12 are essential, whereas Cdc10 and Cdc11 are not. However, the deletion of *CDC10* and *CDC11* leads to defects in cytokinesis. During hyphal growth, Cdc11 is phosphorylated by the cyclin-CDK (cyclin-dependent kinase) complex Ccn1-Cdc28, and another cyclin-CDK complex Hgc1-Cdc28 maintains its phosphorylated state; mutations to the phosphorylation sites in Cdc11 impair the maintenance of polarized growth [176]. Cdc11 phosphorylation by the septin ring-associated kinase Gin4 primes it for further phosphorylation by Ccn1-Cdc28 [176]. Both *cdc10*Δ/Δ and *cdc11*Δ/Δ mutants have abnormalities in septum formation during hyphal growth and form curved hyphae [177,178]. Cdc10 dynamics are dependent on Sep7 and its phosphorylation status [175]. *sep7*Δ/Δ mutants can form hyphae, but the hyphal compartments separate after cytokinesis. Ccn1-Cdc28 and Hgc1-Cdc28 phosphorylate Gin4, which in turn phosphorylates Sep7 [179,180]. Deletion of *GIN4* disrupts the formation of septin rings in germ tubes resulting in a severe cytokinesis defect; *gin4*Δ/Δ mutants form pseudohyphae constitutively and cannot form true hyphae upon serum induction [181]. Depletion of Gin4 in G1 cells blocks septin ring formation [180]. Sep7 is dephosphorylated by the protein phosphatase 2A (PP2A), mediated by the structural subunit Tpd3 and the catalytic subunit Pph21 [182]. Deletion of *PPH21* or *TPD3* or its regulators, *CDC55* or *RTS1*, leads to the hyperphosphorylation of Sep7 and the disruption of septin organization [182,183]. *cdc55*Δ/Δ mutants grow as pseudohyphae under yeast growth conditions, while *rts1*Δ/Δ mutants grow as round, enlarged multinucleated cells. Both *cdc55*Δ/Δ and *rts1*Δ/Δ mutants display hyphal defects.

The nucleus migrates out from the mother cell to the septin band within the developing hyphae, and the first nuclear division occurs in this subapical compartment [184]. One daughter nucleus migrates back to the mother cell, while the other nucleus migrates to the apical compartment. After mitosis, the protein phosphatase Cdc14, which regulates mitotic exit, localizes to the septum in yeast cells and dephosphorylates the Mob2-Cbk1 complex, allowing the transcription factor Ace2 to translocate to the nucleus and activate the transcription of genes involved in cell separation [185]. However, in hyphal cells, Cdc14 does not localize to the septum, and Mob2-Cbk1 remains at the hyphal tip [185]; thus, cytokinesis does not result in cell separation or the formation of a constriction between cells as observed in yeast or pseudohyphae, respectively. The septin ring splits into two rings with the formation of the primary septum dividing the hyphal compartments. Both rings are maintained in hyphal cells, unlike in yeast and pseudohyphal cells, where the septin rings are dissembled after cytokinesis. However, in *sep7*Δ/Δ mutants, Cdc14 can localize to the hyphal septum, activating the Ace2-dependent cell separation program, resulting in hyphal cell separation [186]. The subapical compartment of the hyphae is vacuolated and remains in the G1 phase.

The nucleosome assembly protein, Nap1, plays a role in septin ring formation and dynamics [187]. Deletion of *NAP1* results in constitutively pseudohyphal cells that can transit to true hyphae under hyphal-inducing conditions. Phosphorylation of Nap1 occurs in a cell cycle-dependent manner, which involves Gin4 and Cla4, a second septin ring-associated kinase. Phosphorylated Nap1 translocates from the cytoplasm to the emerging bud neck. In *cdc10*Δ/Δ and *cdc11*Δ/Δ mutants, Nap1 remains in the cytoplasm even though it is hyperphosphorylated. After mitosis, Nap1 is dephosphorylated in a manner that is dependent upon PP2A and Cdc14.

### 6.2. Polarization of the Actin Cytoskeleton

Actin cytoskeleton polarization is required for the morphogenesis of *C. albicans*, regardless of cell type. The actin cytoskeleton, made up of actin patches and cables, maintains directional growth by directing vesicular flow for tip expansion. In yeast and pseudohyphae, polarized growth is driven by the polarisome, a complex that includes the polarisome scaffold protein Spa2, the formin Bni1 that serves as the actin cable nucleator, and the formin-actin-binding protein Bud6 [188]. Actin cables, comprised of long bundles of actin filaments, converge at the apical site. During polarized growth, post-Golgi membrane-bound secretory vesicles are continuously delivered to the apical site, supplying material required to expand the plasma membrane and synthesize new cell walls. The vesicles are tethered to the actin cables by the Rab-type GTPase Sec4, activated by the GEF Sec2 [189,190], while the class V myosin, Myo2, complexed to the regulatory light chain Mlc1, provides the motive force for vesicle transport [191]. Upon arrival at the plasma membrane, the secretory vesicles dock with the exocyst before fusing with the plasma membrane. The exocyst is a complex that comprises Sec3, Sec5, Sec6, Sec8, Sec10, Sec15, Exo70, and Exo84 [192]. Sec4 mediates vesicle tethering with the exocyst through its interaction with Sec15 [189,190].

Although the polarisome and exocyst complexes also localize to the hyphal tip, polarized growth in hyphae is driven by a Spitzenkörper, a vesicle-rich structure responsible for hyphal growth directionality, which is present during all stages of the cell cycle, including septation [193]. Spa2, Bni1, and Bud6 coordinate the functions of the Spitzenkörper and the polarisome complex at the hyphal tip [191,193]. During hyphal growth, the post-Golgi secretory vesicles travel along actin cables to the Spitzenkörper, which acts as a vesicle supply center and is maintained at a fixed distance from the hyphal tip (Figure 5). The vesicle-associated proteins Sec4, Sec2, and Mlc1 are localized to the Spitzenkörper during hyphal growth [191,194]. At the Spitzenkörper, the secretory vesicles are loaded onto actin cables nucleated by the polarisome and transported to the plasma membrane, where they dock with the exocyst. Actin cables are essential in hyphal growth, as their disruption inhibits hyphal formation [193]. Loss of *BNI1* does not affect bud emergence, as germ tube formation can be initiated in *bni1*Δ/Δ mutants. However, the germ tubes are wider in diameter, and *bni1*Δ/Δ mutants cannot maintain polarized cell growth [195]. Deletion of *SPA2* leads to polarity and hyphal growth defects [196]. *spa2*Δ/Δ mutants display random budding with multiple surface protrusions. Similar to the *bni1*Δ/Δ mutants, *spa2*Δ/Δ mutants can form germ tubes. However, unlike in *bni1*Δ/Δ mutants, hyphal growth can be maintained in the *spa2*Δ/Δ mutants, albeit in the form of severely swollen and curvy hyphae. Actin depolymerizing drugs, cytochalasin A and latrunculin A, disrupt the actin cytoskeleton, thus inhibiting hyphal growth and also suppressing the expression of hyphal-specific genes [143,197,198]. Chlorpropham, a drug affecting actin microfilament organization, inhibits hyphal growth [199].

The extensive exocytosis, which occurs at the apical tip and allows for rapid cell wall and membrane deposition, is counterbalanced by endocytosis. Endocytosis is essential for hyphal growth. Suppression of endocytosis suppresses hyphal elongation, and inhibition of endocytosis blocks hyphal formation, while yeast proliferation is unimpeded in both situations. Actin patches form the sites of endocytosis, which is important for maintaining polarity through the endocytic recycling of polarity proteins [200,201]. Before budding or germ tube evagination, cortical actin patches cluster at the apical site [197,202]. Actin patches are highly dynamic, with a lifetime of 5–20 s [203]. As the bud continues to enlarge in yeast cells, the cortical actin patches are redistributed isotropically throughout the bud surface [202]. However, in hyphal cells, the cortical actin patches remain clustered at the hyphal tip throughout hyphal growth [202]. Endocytosis in *C. albicans* mainly occurs via clathrin-mediated endocytosis, and various genes involved in the process have been studied. Sla1 and Sla2 are actin cytoskeletal proteins involved in actin patch organization and dynamics, as well as actin cable polarization, and necessary for normal endocytosis [204,205,206,207,208,209]. Cortical actin patches formed in *sla1*Δ/Δ mutants are depolarized and less dynamic and form short filaments [206,210]. *sla2*Δ/Δ mutants cannot undergo hyphal and pseudohyphal growth as the localization and orientation of actin patches and cables are defective [204,205]. *sla2*Δ/Δ mutants grow slower and form enlarged cells, as Swe1, the morphogenesis checkpoint kinase, delays cell cycle progression. Swe1 phosphorylates the Clb2-Cdc28 complex in response to perturbations to the actin cytoskeleton, thus delaying the normal transition from polarized growth to isotropic bud growth and delaying nuclear division. Pan1 is a clathrin-mediated endocytosis scaffold protein that is essential for endocytosis [211]. Depletion of Pan1 leads to the formation of thick and swollen cells that have abnormal filamentation. The inhibitory protein kinase Akl1 interacts with Pan1 to repress endocytosis, suppressing hyphal elongation [212]. Deletion of *AKL1* results in faster hyphal elongation rates and longer hyphae, while *AKL1* overexpression reduces hyphal elongation rates. However, overexpression of *PAN1* counteracts the effects of *AKL1* overexpression.

The myosin type I protein Myo5, the Wiskott–Aldrich Syndrome protein (WASP) homolog *WAL1*, and the WASP-interacting protein Vpr1 form a complex similar to that in *Saccharomyces cerevisiae* [213]. The Vpr1-Wal1-Myo5 complex is required for the polarized distribution of cortical actin patches. The deletion of *MYO5* leads to mislocalization of cortical actin patches, with the patches dispersed throughout the bud and the mother cell, resulting in excessive isotropic growth [214]. *myo5*Δ/Δ mutants are unable to endocytose and cannot form hyphae [214]. Deletion of *WAL1* and *VRP1* leads to defects in polarized growth [213,215]. *wal1*Δ/Δ mutants can initiate but cannot maintain hyphal growth. Instead, *wal1*Δ/Δ mutants form elongated, pseudohyphal cells under hyphae-inducing conditions. *vpr1*Δ/Δ mutants have a defect in hyphal formation that is slightly less severe than in *wal1*Δ/Δ mutants. Cortical actin patches are depolarized in both the mother cells and buds of *vpr1*Δ/Δ mutants. Myo5 and Wal1 activate the actin module Arp2/3 complex to initiate actin polymerization. Deleting *ARP2* or *ARP3* leads to an inability to form hyphae, although endocytosis is not abolished [208,209]. Deleting *RVS161* and *RVS167*, which encode Bin-Amphiphysin-Rvs (BAR) domain proteins, results in defective actin patch polarization, with the *rvs161*Δ/Δ mutants displaying a more severe defect in endocytosis and morphogenesis than the *rvs167*Δ/Δ mutants [207].

### 6.3. The Role of Ras- and Rho-Family GTPase

The small Ras- and Rho-family GTPases play essential roles in hyphal maintenance. The small Rho GTPase Cdc42 is the master regulator of polarized growth. Cdc42 affects hyphal growth and maintenance in at least two ways. Firstly, Cdc42 affects morphogenesis at the transcriptional level. Reduced expression levels of Cdc42 lead to decreased expression of hyphal-specific genes [216]. Secondly, decreasing cellular levels of active Cdc42 results in yeast and hyphae larger and rounder in shape, indicative of polarized growth defect [216,217]. Cdc42 cycles between GDP- and GTP-bound states. The GEF Cdc24 mediates the formation of GTP-bound Cdc42 [142,216]. [142,216]. The GAPs Rga2 and Bem3 mediate the return of Cdc42 from the GTP- to the GDP-bound form. Cdc42 and Cdc24, both required for viability, localize to the hyphal tip during hyphal growth [142,216]. Bem3 is localized to the apical zone of polarized growth, while Rga2 is localized to the septum and is phosphorylated in a hyphal-specific manner [218]. Loss of *RGA2* and *BEM3* results in the formation of a Spitzenkörper-like structure under pseudohyphal-promoting conditions, and the mutants have a morphology resembling true hyphae. Bem1 is a polarity establishment scaffolding protein that binds GTP-bound Cdc42, keeping it localized to the apical site [219].

The Ras-like GTPase Rsr1, a landmark protein that is the master regulator of the bud site selection system, regulates the amount and distribution of Cdc42 activity at the hyphal tip [220,221]. Rsr1 activity is regulated by the GEF Bud5 and the GAP Bud2. Bud5 is localized to the apical site, while Bud2 is localized to the subapical region and septin ring. *rsr1*Δ/Δ mutants have defects in polarized growth; yeast cells are larger and rounder, while the hyphae are wider than wild-type cells. Rsr1 is involved in regulating the recruitment and spatial distribution of vesicles at the hyphal tip [220,221]. Loss of Rsr1 affects the size of the fixed region to which vesicles are delivered and also affects the localization of exocyst subunits [222]. Rsr1 may play a role in limiting the competition for Cdc42 between the septum and the hyphal tip.

## 7. CDKs, Cyclins, and Their Roles in Hyphal Morphogenesis

Maintenance of cell signaling is important for cell cycle progression and cell growth. The cell cycle-associated cyclins and CDKs tightly regulate the small GTPases and other components of polarized growth. *C. albicans* has three G1 cyclins (Ccn1, Cln3, and Hgc1) and two B-type mitotic cyclins (Clb2, Clb4), of which only Cln3 and Clb2 are essential. The essential CDK Cdc28 serves as the master regulator that controls cell cycle progression at G1/S and G2/M phases via specific cyclin interactions that dictate the timing of the phases. Levels of the G1 and B-type mitotic cyclins oscillate during the cell cycle, and a single cyclin-Cdc28 complex can regulate multiple events within each phase of the cell cycle. Cdc28 is usually stable and present at constant levels throughout the cell cycle; however, its depletion leads to filamentous growth [223]. Ccn1 and Cln3 levels in yeast cells are high in the G1 phase, coinciding with bud emergence and apical growth, and decline in the early G2 phase. Clb2 levels peak in the early G2/M phase, while Clb4 levels reach their peak in the mid-G2/M phase [224]. Levels of both B-cyclins start to decline in the M phase and disappear during exit from mitosis [185,224].

In hyphal cells, polarized growth continues at the apical site throughout the cell cycle, indicating the decoupling of cell elongation from the cell cycle. Ccn1 and Cln3 levels are accumulated earlier and persist for a longer time during hyphal growth [224,225], extending the G1 phase in the early germ tube. Accumulation of the mitotic cyclins, Clb2 and Clb4, is delayed in hyphal cells. Although it is not required for the initiation of hyphal growth, high levels of Ccn1 are required for maintenance of hyphal growth, along with Cln3. The forkhead family transcription factor, Fkh2, usually undergoes cell cycle-dependent phosphorylation to induce the expression of genes that regulate cell cycle progression [226,227]. However, upon hyphal induction, Fkh2 is phosphorylated by Ccn1/Cln3-Cdc28 and Mob2-Cbk1 in a cell cycle-independent manner, redirecting it to enhance the expression of hyphal-specific genes such as the hyphal-specific G1 cyclin *HGC1* (Figure 6) [226,227]. *fkh2*Δ/Δ mutants grow constitutively as pseudohyphae under both yeast and hyphal-inducing conditions [226,227]. During hyphal growth, Ccn1-Cdc28 and Cln3-Cdc28 complexes phosphorylate Mob2, the activator of Cbk1, the cell wall integrity kinase, inhibiting the activation of Ace2 (Figure 6) [228]. Cln3-Cdc28 complex regulates cortical actin patches via phosphorylation of Sla1 [206].

The hyphal-specific G1 cyclin Hgc1 does not regulate the cell cycle but plays a critical role in hyphal morphogenesis (Figure 6) [169]. Besides suppression by Tup1 and Nrg1, the expression of *HGC1* is positively regulated by the transcription factor Ume6, which ensures that Hgc1 is expressed throughout the cell cycle as long as the inducing conditions remain [169]. Hgc1 interacts with Cdc28, forming a complex regulating by phosphorylation regulators and components of cell polarity, membrane trafficking, and cell separation, which is required to maintain hyphal growth (Figure 6). The Hgc1-Cdc28 complex phosphorylates and inactivates Rga2, sequestering it from the hyphal tip to allow Cdc42 localization at the hyphal tip to persist during polarized growth [217,229]. Hgc1-Cdc28, together with Clb2-Cdc28, phosphorylates Spa2, localizing the polarisome to the hyphal tip [230]. Hgc1-Cdc28 complex phosphorylates the exocyst subunits Exo84 and Sec2, allowing them to be recycled at the growing hyphal tip [194,231]. The Hgc1-Cdc28 complex phosphorylates Efg1, leading to Efg1 competitively binding to promoters of Ace2 target genes, thereby repressing the expression of cell separation activators to prevent cell separation after cytokinesis [232]. The Hgc1-Cdc28 complex also plays a role in regulating the septin ring dynamics during hyphal growth via Sep7 [175].

The cyclins Pcl1 and Pcl5 and the CDK Pho85, although not essential for cell cycle progression, contribute to morphogenesis in response to environmental cues. The Pcl1-Pho85 complex is required for temperature-dependent filamentation induced by Hsp90 inhibition [233]. The transcription factor Hms1 is required for filamentation induced by high temperatures. The Pcl1-Pho85 complex phosphorylates Hms1, which then binds to hyphal-specific genes. It also regulates the degradation of the transcription factor Gcn4, which is indirectly involved in filamentous growth in response to amino acid starvation [234]. Gcn4 induces *PCL5* expression, and the Pcl5-Pho85 complex phosphorylates Gcn4, leading to its degradation [235,236].

## 8. Cell Cycle Perturbation Leads to Morphogenesis

Although hyphal growth is not directly controlled by the cell cycle, perturbing the cell cycle can cause significant pseudohyphal growth under non-hyphal-inducing conditions or block hyphal growth under hyphal-inducing conditions [134,135,237]. Loss of Ccn1 does not induce morphogenesis but causes a filamentation defect under serum induction [225]. Depletion of Cln3, Clb2, or Clb4 results in filamentous growth in the absence of hyphal-inducing stimuli [134,237]. Cells depleted of Cln3 undergo cell cycle arrest in the G1 phase and form filaments before the resumption of the cell cycle [135]. In the absence of mitotic cyclins, polarized growth promoted by G1 cyclins is not entirely suppressed, and filamentation occurs. Clb2-depleted cells form elongated projections during cell cycle arrest and are inviable, whereas Clb4-depleted cells grow constitutively as pseudohyphae and remain viable [224]. Depletion of Cdc28 also promotes filamentous growth [223].

Genotoxic stresses that disrupt cell cycle progression and activate DNA damage/replication checkpoints lead to filamentation [238]. Pharmacological inhibition of cell cycle progression by the DNA replication inhibitors hydroxyurea (HU) and aphidicolin (AC) or DNA damage induced by UV radiation or the alkylating agent methylmethane sulfonate (MMS) result in S phase arrest, inducing filamentous growth [133,239,240]. Checkpoint proteins play a crucial role in response to DNA replication and DNA damage stresses. The protein kinase Rad53 plays a central role in the DNA replication and DNA damage checkpoint. *rad53*Δ/Δ mutants cannot switch to filamentous growth in response to DNA replication inhibitors and DNA damage; mutations to the Rad53 FHA domains inhibit filamentation in response to DNA damage, but not cell cycle arrest [241,242]. Deletion of *RAD9*, which encodes a checkpoint protein upstream of Rad53, blocks DNA damage-induced filamentation [241]. Depleting the DNA repair protein Rad52 or deleting its gene results in the accumulation of spontaneous DNA damages that trigger the DNA damage checkpoint, resulting in filamentous growth [243]. After the stress is relieved, deactivation of the cell cycle checkpoint is necessary for the cell cycle to progress. Rad53 is dephosphorylated by the protein phosphatase 2A-like complex Pph3-Psy2. Deletion of *PPH3* that encodes the catalytic subunit, *PSY2* that encodes the regulatory subunit, or *TIP41* that encodes the regulator of the Pph3-Psy2 complex enhances MMS-induced filamentous growth and delays the filament-to-yeast transition following DNA damage stress relief [239,240]. The histone acetyltransferases Hat1 and Hat2 are required for the repair of DNA damages caused by endogenous and exogenous agents; *hat1*Δ/Δ mutants accumulate DNA damages rapidly and switch to filamentous growth [244].

Perturbations to mitosis can also lead to the switch to filamentous growth under non-hyphal-inducing conditions. The cell cycle regulatory polo-like kinase Cdc5 is required for the early stages of nuclear division and chromatin separation and mediates spindle formation during the S phase [245]. Depletion of Cdc5 leads to mitotic inhibition and blocks the cell cycle in the G2 phase, leading to hyphal-like growth; however, the cells eventually lose viability [245]. The cytoplasmic dynein, Dyn1, mediates nuclear movement during mitosis. Deletion of *DYN1* or depletion of Dyn1 results in filamentous growth, which requires the spindle position checkpoint protein Bub2 [246,247]. Pharmacological perturbation of mitosis by the microtubule inhibitor nocodazole and deletion of *MAD2* that encodes a spindle assembly checkpoint protein leads to pseudohyphal growth [248].

Degradation of cyclins and CDK inhibitors is regulated mainly by ubiquitin-proteasome-dependent proteolysis and is required for orderly cell cycle progression. Degradation of these cell cycle regulatory proteins is mediated by two multiprotein ubiquitin ligase E3 complexes, the Skp1-Cullin/Cdc53-F-box (SCF) complex and the anaphase-promoting complex/cyclosome (APC/C) complex. The multiprotein SCF complex, consisting of the linker protein Skp1, the scaffold protein Cullin/Cdc53, and a substrate recognition F-box protein, plays a central role in regulating the temporal and spatial degradation of cell cycle regulatory proteins. Depleting the essential *CDC53* or the deletion of the F-box protein genes *CDC4* and *GRR1* leads to filamentous growth. SCF^Cdc4^ is required for the degradation of the CDK inhibitor Sol1 and the transcription factors Ume6 and Gcn4 [235,236,249]. Deleting *CDC4* leads to constitutively filamentous growth with a mix of hyphal and pseudohyphal cells [250]. As Sol1 represses the Clb2-Cdc28 complex, deletion of *CDC4* stabilizes Sol1, inhibiting the switch from apical to isotropic growth, resulting in a pseudohyphal phenotype. However, deleting *SOL1* in the *cdc4*Δ/Δ mutant background still gives rise to constitutively filamentous growth [249]. SCF^Grr1^ is required for the degradation of Ccn1 and Cln3. The deletion of *GRR1* stabilizes Ccn1 and Cln3 levels, and the *grr1*Δ/Δ mutants grow constitutively as pseudohyphae under yeast conditions [251].

The APC/C complex mediates protein degradation during mitotic progression [252]. While little is known about the APC/C complex in *C. albicans*, the two co-activators, Cdc20 and Cdh1, have been identified recently. Cdc20 is essential and mediates the degradation of Clb2 and Cdc5. Depletion of Cdc20 results in the accumulation of Clb2 and Cdc5, leading to a delay in metaphase and telophase; Cdc20-depleted cells grow as long filaments over time [252]. Cdh1 likely plays a role in regulating mitotic exit by influencing Clb2 and Cdc5 degradation; deletion of *CDH1* results in a delay in Clb2 degradation and elevated levels of Cdc5 [252]. *cdh1*Δ/Δ mutants display pleiotropic phenotypes, with a mix of yeast, elongated buds, and pseudohyphae.

## 9. Conclusions

In summary, the extensive research findings over the years have provided us with illuminating insights into the activation and regulation of hyphal morphogenesis in *C. albicans*. The factors involved are often crucial in controlling the balance between commensalism and invasive infection by *C. albicans*. The yeast-to-hyphal transition in *C. albicans* is highly dependent on the complex interplay between internal signal transduction signaling pathways and external environmental cues that reflect the host niches. This transition is governed by a complex network of signaling pathways, including the cAMP-PKA pathway, the MAP kinase pathways, and the Cek1-mediated and PKC pathways. Activation of these pathways in response to various cues triggers the activation of specific transcription factors such as Efg1, Flo8, Ume6, Tec1, and Cph1. Crosstalk between the cAMP-PKA and MAP kinase pathways add a further layer of complexity to the existing signaling network as multiple signaling pathways can converge to the same set of transcription factors. Following hyphal initiation, subsequent hyphal development requires delicate mechanisms to maintain hyphal elongation. Polarized growth requires continuous delivery of membrane-bound secretory vesicles, along the actin cables, to the site of polarized growth. The vesicles accumulate as Spitzenkörper in the subapical region before docking with the exocyst and polarisome components. This exocytosis process is counterbalanced by the endocytosis process. Cell morphology of *C. albicans* is known to be tightly linked to cell cycle progression through cyclin-CDK complexes. One of the most important CDK complexes is the Hgc1-Cdc28 complex, which governs multiple cellular processes required for hyphal development. The Hgc1-Cdc28 complex plays important roles in polarized growth, polarized secretion, and inhibition of cell separation, which ensures the formation of long tubular cells without constriction at the septal junction. Lastly, perturbation of the cell cycle can either induce or impair the highly polarized growth in *C. albicans* under different conditions.

## 10. Future Directions

However, much remains to be explored and unraveled in *C. albicans* morphogenesis. While the mechanisms behind hyphal induction during cell cycle arrest have been uncovered, there are still missing gaps. Future work could uncover the genes that regulate filamentation in response to the cell cycle perturbation and elucidate how the signals are transduced to the cAMP-PKA pathway and the activated downstream transcriptional regulators. The link between nutrient and osmotic stress and filamentous growth via the PKC pathway has been uncovered, but the downstream transcriptional regulators remain elusive. Applying evolutionary tools of systems biology in combination with animal-based studies will propel discoveries in this field. The recent development of the transposon-mediated mutagenesis systems in haploid *C. albicans* strains would allow genome-wide screening for novel genes with functions that influence morphogenesis [253,254]. Future work towards identifying additional downstream transcriptional regulatory genes could open up new avenues towards antifungal therapy development.

## Figures and Tables

**Figure 1 pathogens-10-00859-f001:**
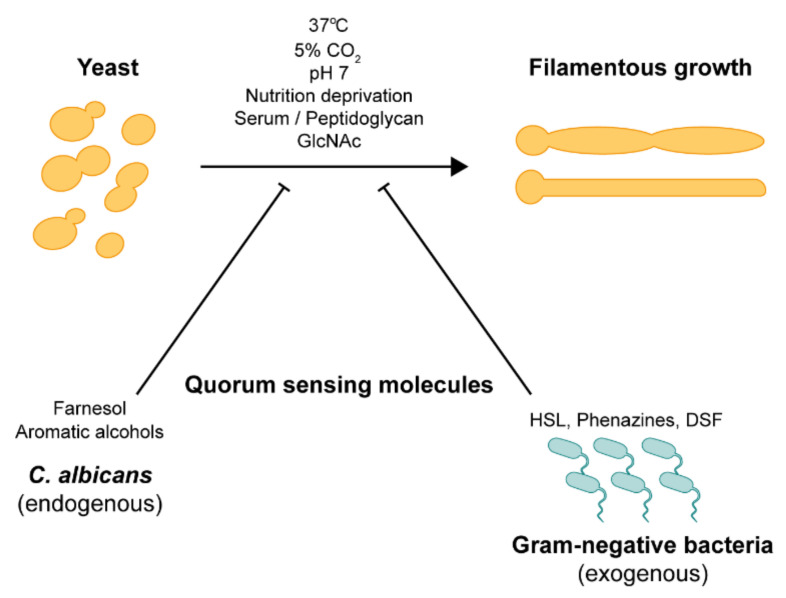
External hyphal-inducing signals. The yeast-to-hyphae transition in *C. albicans* can be triggered by various environmental cues such as high temperature (37 °C), high CO_2_ concentration (~5%), pH 7, nutrition deprivation, serum, peptidoglycan, *N*-acetylglucosamine, and inhibited by quorum-sensing molecules from endogenous and exogenous sources.

**Figure 2 pathogens-10-00859-f002:**
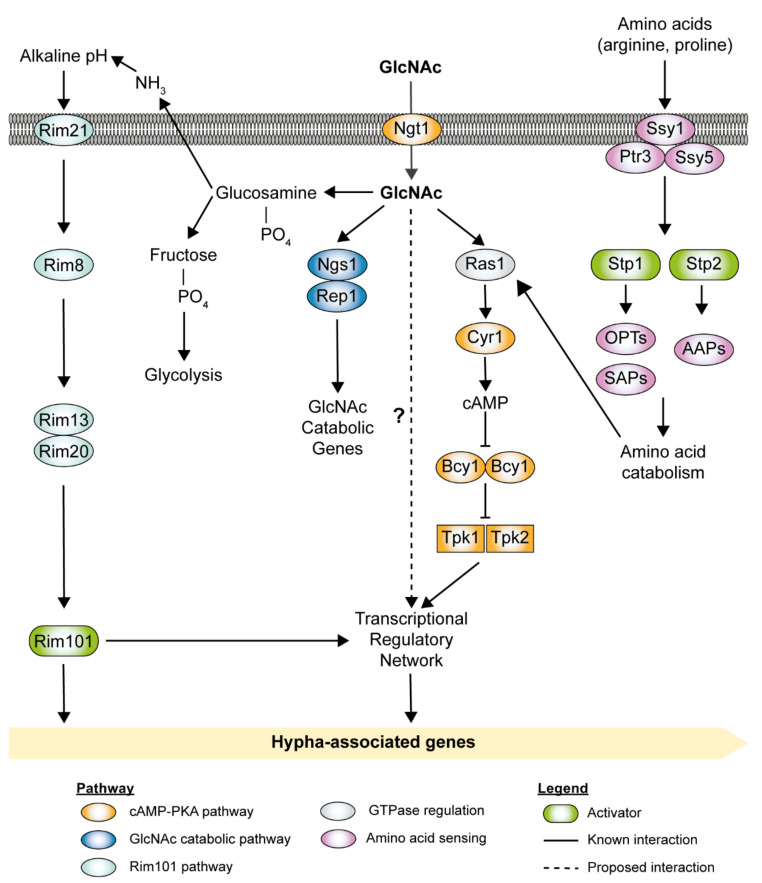
*N*-acetylglucosamine (GlcNAc) and amino acid-induced signal transduction pathways in *C. albicans*. Ngt1, localized in the plasma membrane, transports GlcNAc into the cell. However, when GlcNAc is present in high concentrations, it can enter the cell through diffusion. The main signal transduction pathway for GlcNAc-induced hyphal growth was initially thought to be the cAMP-PKA pathway. Recently, the transcription factors Ngs1 and Rep1, which are involved in GlcNAc catabolism, were found to stimulate hyphal growth via a cAMP-independent pathway. GlcNAc catabolism also increases the extracellular pH, which favors the hyphal growth via the alternate Rim101 pathway. Extracellular amino acids are detected via the plasma membrane-localized SPS (Ssy1-Ptr3-Ssy5) complex. The SPS-sensor activation leads to endoproteolytic processing at the nuclear exclusion domain of transcription factors Stp1 and Stp2. Processed Stp1 regulates the expression of secreted aspartyl proteinase (e.g., *SAP2*) and oligopeptide transporters (e.g., *OPT1* and *OPT3*), while processed Stp2 regulates the expression of amino acid permeases (APPs).

**Figure 3 pathogens-10-00859-f003:**
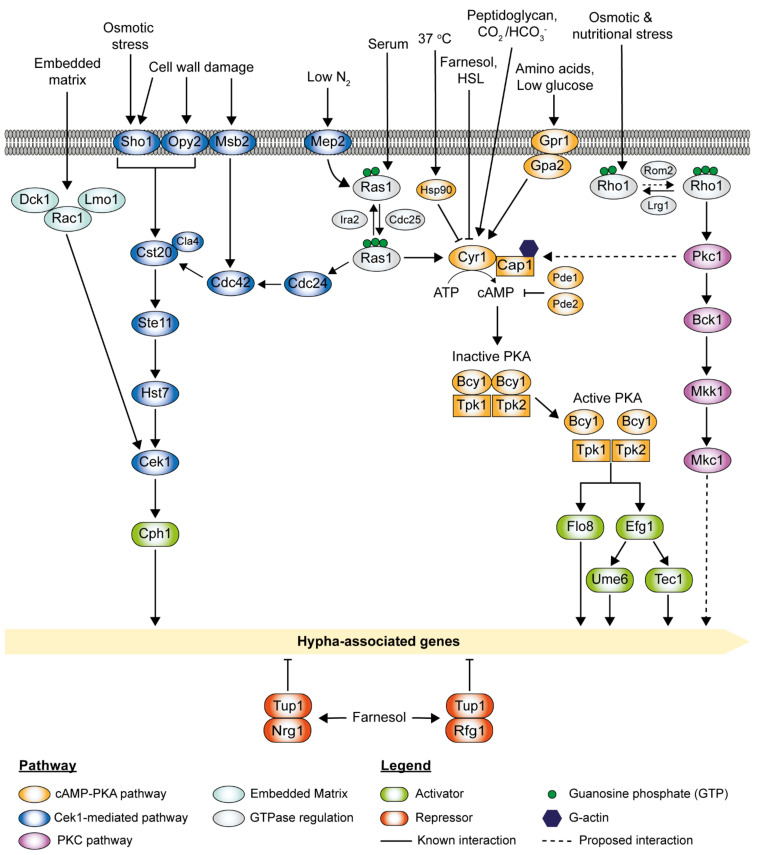
Signal transduction pathways that govern hyphal growth in *C. albicans*. Activation of filamentous growth in *C. albicans* by various environmental cues and signal transduction pathways; the cAMP-PKA pathway, the Cek1-mediated pathway, the PKC pathway, and the embedded matrix.

**Figure 4 pathogens-10-00859-f004:**
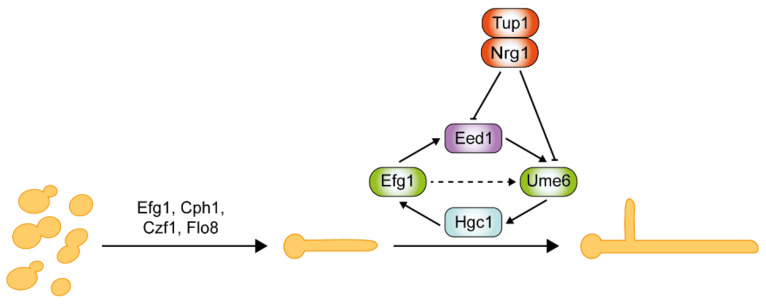
**Regulation of hyphal elongation requires mechanisms for initiation and long-term maintenance.** Initiation of hypha growth requires transcription factors such as Efg1, Cph1, Czf1, and Flo8. Subsequent elongation process and maintenance require the involvement of Hgc1, Eed1, and Ume6. Both Eed1 and Ume6 are negatively regulated by Tup1 and Nrg1.

**Figure 5 pathogens-10-00859-f005:**
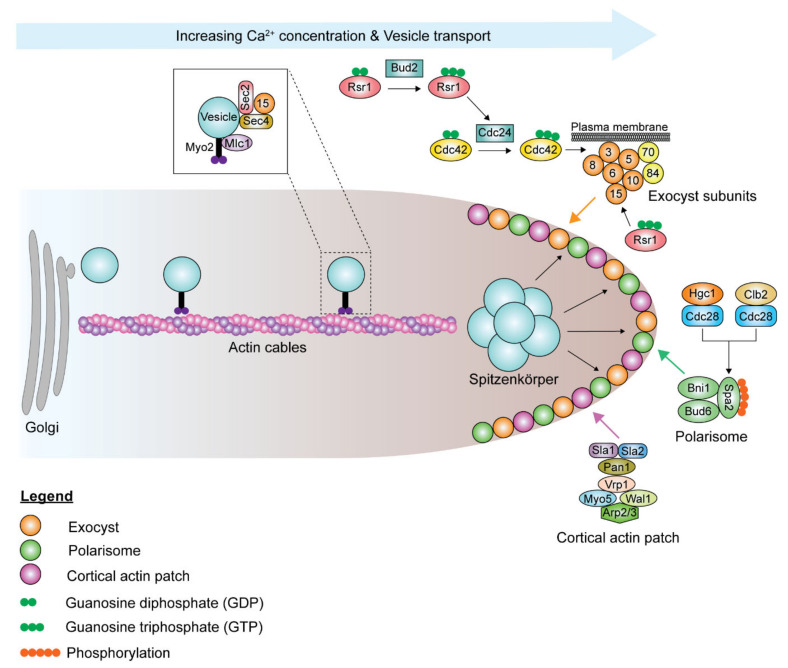
Schematic representation of polarized growth in *C. albicans* hyphal cells. Polarized growth is driven by Spitzenkörper, a vesicle supply center maintained at a fixed distance from the hyphal tip. Post-Golgi membrane-bound secretory vesicles are continuously delivered to the site of polarized growth. Secretory vesicles, tethered by the Rab-type GTPase Sec4 and the GEF Sec2, are transported to the hyphal tip via actin cables with the class V myosin Myo2 complexed to the regulatory light chain Mlc1, providing the motive force. The vesicles accumulate in Spitzenkörper before docking with the exocyst, which consists of Sec3, Sec5, Sec6, Sec8, Sec10, Sec15, Exo70, and Exo84 subunits, before fusing with the plasma membrane. Spa2, Bni1, and Bud6 coordinate the functions of the Spitzenkörper and the polarisome complex at the apical site of the hyphal tip. Endocytosis, endocytic recycling of polarity proteins, involves the cortical actin patches at the apical site of the hyphal tip. Actin patch organization and dynamics involve the actin cytoskeletal proteins Sla1 and Sla2, the actin skeleton-regulatory protein Pan1, and the Vpr1-Wal1-Myo5 complex, which activates the Arp2/3 complex. The landmark GTPase Rsr1, upon activation by its GEF Bud2, localizes Cdc24 to the site of tip growth, in addition to Ca^2+^ binding of the EF-hand motif in Cdc24.

**Figure 6 pathogens-10-00859-f006:**
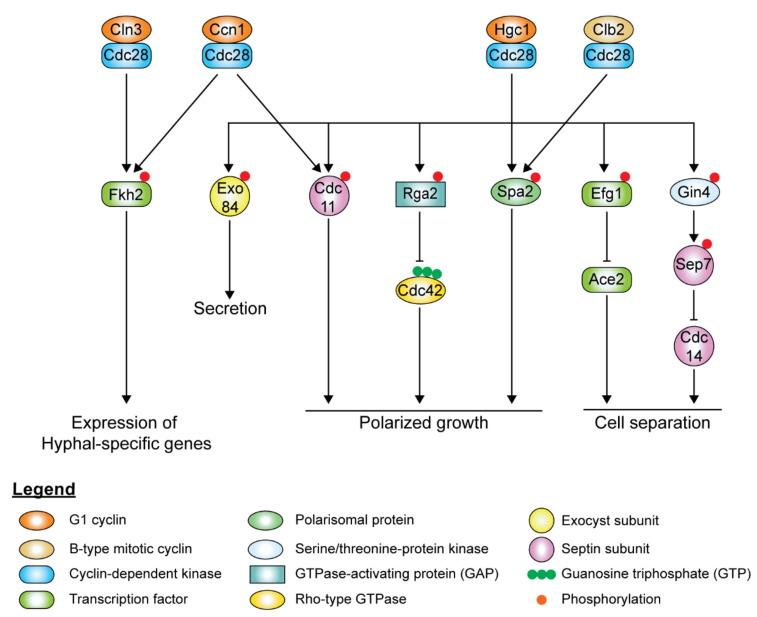
*C. albicans* morphogenesis is tightly regulated by the cell cycle-associated cyclins and cyclin-dependent kinase (CDK). The essential CDK Cdc28 serves as the master regulator that controls the cell cycle progression at G1/S and G2/M phases by forming CDK complexes with specific cyclins. Levels of Cdc28 are relatively stable throughout the cell cycle and deplete during hyphal growth. In contrast, levels of the G1 and B-type mitotic cyclins oscillate during the cell cycle. G1 cyclins Cln3 and Ccn1 peak in the G1 phase and decline in the early G2 phase, while B-type mitotic cyclin Clb2 peaks in the early G2/M phase and declines in the M phase. Upon hyphal induction, Fkh2 is phosphorylated by Cln3-Cdc28 and Ccn1-Cdc28 complexes in a cell cycle-dependent manner to enhance the expression of hyphal-specific genes. The Hgc1-Cdc28 complex is essential for the maintenance of hyphal growth. The exocyst subunit Exo84 is phosphorylated by the Hgc1-Cdc28 complex for the regulation of polarized secretion. Phosphorylation of the septin subunit Cdc11 (by Ccn1-Cdc28 and Hgc1-Cdc28 complexes), GAP Rga2 (by the Hgc1-Cdc28 complex), and the polarisome protein Spa2 (by Hgc1-Cdc28 and Clb2-Cdc28 complexes) promote polarized growth. Rga2 is phosphorylated and inactivated by Hgc1-Cdc28, which relives the repression of the GTPase Cdc42. Phosphorylation of the transcription factor Efg1 and protein kinase Gin4 inhibit cell separation. Phosphorylated Efg1 binds to promoters of Ace2 target genes, inhibiting their transcription. Phosphorylated Gin4 modifies the dynamics of the septin ring by subsequent phosphorylation of Sep7 and deactivating the cell separation program via inhibition of the protein phosphatase Cdc14.

## Data Availability

Not applicable.

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
