# Peer review of "From Jekyll to Hyde: The Yeast–Hyphal Transition of *Candida albicans"

_pathogens, 2021, doi:10.3390/pathogens10070859_

Round 1

Reviewer 1 Report

Excellent and concise review on yeast to hyphae transition and how the process is initiated and regulated.

Correct 35C on line 115

Author Response

Reviewer 1: Excellent and concise review on yeast to hyphae transition and how the process is initiated and regulated. Correct 35C on line 115.
Author's Response: We thank the reviewer for acknowledging our efforts in this review preparation. The typo error has been rectified. Please refer to Page 3, line 116.

Reviewer 2 Report

The manuscript “From Jekyll to Hyde: The yeast-hyphal transition of Candida albicans” presents a well-updated account of the effect of environmental and genetical factors on yeast-to-hyphae transition. It is a well-written manuscript, balanced and clear. The figures are informative and easy-to-follow. The references current and up-to-date. The comprehensive reviews like this one may contribute to developing alternative therapies, which based on the disruption of the hyphae formation.

Minor Comments:

I noticed some spelling mistakes; however, they are absolutely correctable in the case of proofreading (if this review will be accepted).

Author Response

Reviewer 2: The manuscript “From Jekyll to Hyde: The yeast-hyphal transition of Candida albicans” presents a well-updated account of the effect of environmental and genetical factors on yeast-to-hyphae transition. It is a well-written manuscript, balanced and clear. The figures are informative and easy-to-follow. The references current and up-to-date. The comprehensive reviews like this one may contribute to developing alternative therapies, which based on the disruption of the hyphae formation.
Author's Response: We thank the reviewer for the thoughtful review of our work and kind words.

Reviewer 2: I noticed some spelling mistakes; however, they are absolutely correctable in the case of proofreading (if this review will be accepted).
Author's Response: We apologize for the spelling mistakes. We have proofread the review and rectified the mistakes in the revised version. 

Reviewer 3 Report

The manuscript is a review on the yeast-hyphal transition of Candida albicans. The review is divided into three parts: First an introduction and a summary of environmental cues for the hyphal transition (sections 1 and 2). Then pathways leading to hyphae (sections 3-5) and lastly on the mechanism of the transition (sections 6-8). The manuscript is well written.

Section 5 ends with a paragraph on negative regulators as potential drug targets. It would be very interesting and helpful if the other sections have similar paragraphs on potential drug targets.

Furthermore, I can not find information on cell wall remodelling and the importance of beta1,3-glucanases (e.g. H. Xu, C. J. Nobile and A. Dongari-Bagtzoglou, PLoS One, 2013, 8, e6373)

Minor points:

Figure 1: The QSMs are labeled with bar-headed lines (i.e. inhibition) However, they are described as triggers in the text.

Page 6, lines 2-3: ethyl alcohol, isoamyl alcohol, and the dodecanols are not aromatic alcohols.

Author Response

Reviewer 3: The manuscript is a review on the yeast-hyphal transition of Candida albicans. The review is divided into three parts: First an introduction and a summary of environmental cues for the hyphal transition (sections 1 and 2). Then pathways leading to hyphae (sections 3-5) and lastly on the mechanism of the transition (sections 6-8). The manuscript is well written.
Author's Response: We thank the reviewer for the expert opinions and suggestions. We have provided a point-by-point reply to the comments as below.

Reviewer 3: Section 5 ends with a paragraph on negative regulators as potential drug targets. It would be very interesting and helpful if the other sections have similar paragraphs on potential drug targets.
Author's Response: We appreciate the reviewer’s comment and have considered this inclusion. However, apart from the negative regulators, compounds that target the specific transcription factors or cell cycle-related genes remain elusive. We hope the reviewer will kindly agree with our justification.

Reviewer 3: Furthermore, I can not find information on cell wall remodelling and the importance of beta-1,3-glucanases (e.g. H. Xu, C. J. Nobile and A. Dongari-Bagtzoglou, PLoS One, 2013, 8, e6373).
Author's Response: We appreciate the reviewer’s comment. The role of beta-1,3-glucanases and their involvement in Candida albicans hyphal morphogenesis and the paper by Xu et al. have been added to the quorum sensing section. Please refer to Page 6, lines 223 – 231. We have not covered cell wall remodeling within the scope of our review manuscript, as our focus was more on the signaling pathways and genetic factors involved in the yeast-to-hyphae transition. While cell wall remodeling occurs during morphogenesis, most of the primary literature focusing on this aspect of C. albicans pathogenesis is related to host immune response.

Reviewer 3: Figure 1: The QSMs are labeled with bar-headed lines (i.e. inhibition). However, they are described as triggers in the text.
Author's Response: We thank the reviewer for pointing out the mistake. We have revised the text as suggested. Please refer to Page 2 Line 77.

Reviewer 3: Page 6, lines 2-3: ethyl alcohol, isoamyl alcohol, and the dodecanols are not aromatic alcohols.
Author's Response: We apologize for the mistake. Ethyl alcohol, isoamyl alcohol and dodecanols are aliphatic alcohols. We have rephrased the sentence for clarity. Please refer to Page 6, Lines 199 – 203.